# Characterization of Cell-Bound CA125 on Immune Cell Subtypes of Ovarian Cancer Patients Using a Novel Imaging Platform

**DOI:** 10.3390/cancers13092072

**Published:** 2021-04-25

**Authors:** Germán González, Kornél Lakatos, Jawad Hoballah, Roberta Fritz-Klaus, Lojain Al-Johani, Jeff Brooker, Sinyoung Jeong, Conor L. Evans, Petra Krauledat, Daniel W. Cramer, Robert A. Hoffman, W. Peter Hansen, Manish S. Patankar

**Affiliations:** 1PNP Research Corporation, Drury, MA 01343, USA; petra@pnpresearch.com (P.K.); peter@pnpresearch.com (W.P.H.); 2Brigham and Women’s Hospital, Department of Obstetrics, Gynecology and Reproductive Biology, Boston, MA 02115, USA; klakatos@bwh.harvard.edu (K.L.); dcramer@bwh.harvard.edu (D.W.C.); 3Thorlabs Imaging Systems, Sterling, VA 20166, USA; jhoballah@thorlabs.com (J.H.); jbrooker@thorlabs.com (J.B.); 4Department of Obstetrics and Gynecology, University of Wisconsin Madison, Madison, WI 53706, USA; rfritzklaus@wisc.edu (R.F.-K.); laljohani@wisc.edu (L.A.-J.); 5Massachusetts General Hospital, Wellman Center for Photomedicine, Boston, MA 02114, USA; sjeong4@mgh.harvard.edu (S.J.); evans.conor@mgh.harvard.edu (C.L.E.); 6Private Researcher, Philadelphia, PA 98516, USA; hoffman.ra@gmail.com

**Keywords:** ovarian cancer, CA125, MUC16, Siglec-9, surface plasmon resonance, multiparameter imaging, deep learning, lymphocyte

## Abstract

**Simple Summary:**

High-grade serous ovarian cancer is a fatal disease typically detected at an advanced stage when options for effective treatment are significantly limited. The lack of a screening modality to identify ovarian cancer in its early stage continues to be a major impediment in the management of this malignancy. The serum biomarker CA125, a repeating peptide epitope present in the sialomucin, MUC16, is unsuitable as a screening test. We have demonstrated that immune cells of ovarian cancer patients capture miniscule amounts of CA125 on their cell surface. Here, we report an automated, sensitive, alignment-free microscopy platform to qualitatively and quantitatively assess the low-abundance binding of CA125 to circulating leucocyte subsets. Through a comparison of the CA125 levels on immune cell subsets of ovarian cancer patients versus healthy donors, we demonstrate that our new technique can serve as a novel diagnostic platform for detection and monitoring of ovarian cancer.

**Abstract:**

MUC16, a sialomucin that contains the ovarian cancer biomarker CA125, binds at low abundance to leucocytes via the immune receptor, Siglec-9. Conventional fluorescence-based imaging techniques lack the sensitivity to assess this low-abundance event, prompting us to develop a novel “digital” optical cytometry technique for qualitative and quantitative assessment of CA125 binding to peripheral blood mononuclear cells (PBMC). Plasmonic nanoparticle labeled detection antibody allows assessment of CA125 at the near-single molecule level when bound to specific immune cell lineages that are simultaneously identified using multiparameter fluorescence imaging. Image analysis and deep learning were used to quantify CA125 per each cell lineage. PBMC from treatment naïve ovarian cancer patients (N = 14) showed higher cell surface abundance of CA125 on the aggregate PBMC population as well as on NK (*p* = 0.013), T (*p* < 0.001) and B cells (*p* = 0.024) compared to circulating lymphocytes of healthy donors (N = 7). Differences in CA125 binding to monocytes or NK-T cells between the two cohorts were not significant. There was no correlation between the PBMC-bound and serum levels of CA125, suggesting that these two compartments are not in stoichiometric equilibrium. Understanding where and how subset-specific cell-bound surface CA125 takes place may provide guidance towards a new diagnostic biomarker in ovarian cancer.

## 1. Introduction

CA125 is a repeating protein epitope of the >2 million Da MUC16 molecule [1,2]. This sialomucin is overexpressed by high-grade serous ovarian tumors [3,4]. Although a transmembrane mucin, MUC16 is shed from the cell surface, and as a result, can be detected in serum using antibodies directed against the CA125 epitope [5]. Quantitative levels of serum CA125 have proven useful to define progression-free survival of serous ovarian cancer patients after they have received first-line therapy [6] and in the management of serous ovarian cancer in patients past their cytoreductive surgery and chemotherapy [7,8,9,10].

It has been established that MUC16 binds to peripheral blood mononuclear cells (PBMC) [11] and, in the case of NK cells, this binding blunts its anti-tumor response [12,13,14]. Our previous studies showing MUC16 binding to PBMC relied on the use of flow cytometry. However, successive experiments indicated that it was often difficult to monitor the low levels of MUC16 bound to PBMC using conventional flow cytometry, hindering further research on the use of cell-bound CA125 in clinical practice. 

It is well-understood that for low levels of cell surface antigens, fluorescent labels are masked by the self-fluorescence of the cell interior, fluorescence of other surface markers being investigated in the same experiment and, moreover, they photo bleach. Alternatively, sub-micron size gold plasmonic nanoparticles (PNP) scatter light 1 million times more intensely than the emission from their fluorescent counterparts [15]. This intense signal is caused when the electrons of the gold nanoparticles couple resonantly with the electromagnetic field of the incident light and reradiate scattered light. When imaged by a darkfield microscope, PNP are 13 times brighter than the background from the cell interior, and they do not photo bleach. Sub-micron, antibody-conjugated gold nanoparticles are therefore excellent candidates for labeling and quantifying low-abundance cell surface antigens by light microscopy. Furthermore, fluorescent staining and PNP staining do not interfere with one another [16]; therefore, orthogonal stains can be considered and be used concomitantly for low-abundance antigen staining and traditional cell lineage estimation [17].

Using PNP conjugated to the anti-CA125 antibody, we recently demonstrated in a small patient cohort that cell-bound CA125 is more abundant on PBMC of serous invasive cancer patients than in healthy controls [18]. This proof-of-principle study validated our initial flow cytometry results by confirming binding of CA125 (MUC16) to PBMC using our newly developed and highly sensitive imaging protocol. However, the study did not provide insights into the relative binding of the mucin to specific lineages of the immune cells. This latter point is important, because MUC16 selectively and preferentially binds to immune cells that express Siglec-9 and hence is not detected on all the immune cells in a population. 

To explore the distribution and differential binding of CA125 across subsets of PBMC, we developed a new oil- and calibration-free, fully automated imaging platform that can acquire a large number of fields of view, performing simultaneous quantitative PNP detection, as in [18], and qualitative cell lineaging through multiparameter fluorescence microscopy, as in [17]. Automation was needed to quantify PNP binding in low-abundance cell populations, and was achieved using X-Y stage movement, autofocusing modules and automated cell lineaging software. 

Using the newly developed system and PBMC from 14 serous invasive ovarian cancer patients and 7 healthy controls, we, for the first time, obtained quantitative measurements of the relative abundance of cell-bound CA125 in PBMC subsets. We found clear elevation of CA125 on NK cells, B cells and T cells of treatment-naive serous ovarian cancer patients, but not on monocytes or NK-T cells. No correlation was found between the new measurements and serum CA125, showing that cell-bound CA125 may become a new independent biomarker with clinical implications for ovarian cancer management. 

## 2. Materials and Methods

### 2.1. Samples

Patients (N = 14) with advanced-stage serous ovarian cancer and healthy controls (N = 7) were recruited and consented under approved Institutional Protocols at the University of Wisconsin (Madison) School of Medicine (protocol 2015-1403) and the Dana Farber Brigham and Women’s Cancers Center (protocol 16-255 and 2018P001677). Serum CA125 was available for 12 of the 14 serous invasive cancer patients.

### 2.2. Sample Staining

PBMC from healthy control and serous invasive patients were isolated from fresh blood samples by density gradient centrifugation (Ficoll-Paque) and cryopreserved in 1 mL vials at a cell density ranging between 3 × 10^6^ and 5 × 10^6^ cells/mL. Cells were separately stained with fluorescent dyes and anti-CA125 conjugated to gold nanoparticles (anti-CA125-PNP) in a two-step process with the following reagents and protocol.

Eighty nm gold nanoparticles conjugated with anti-CA125 were purchased from BBI Solutions (Cardiff, UK) and lyophilized for long-term storage. Fluorescent reagents were bought from Biolegend (San Diego, CA, USA). They were BV510 anti-human CD45, BV421 anti-human CD19, AF488 anti-human CD14, PE anti-human CD56 NCAM and AF647 anti-human CD3 antibody. 

Frozen PBMC were thawed in a 37 °C water bath for less than 1 min until only a small amount of ice was left in the vial. Cells were transferred to 15 mL centrifuge tubes containing 9 mL of pre-warmed 37 °C RPMI with 5% BSA. After mixing, cells were centrifuged at 300× *g* for 10 min and resuspended to a density of 1 × 10^7^ cells/mL in phosphate buffered saline (PBS) with 5% BSA. Fluorescent staining was performed by adding 5 µL of each of the fluorescent dyes and incubating for 30 min at room temperature under continuous mixing conditions (750 rpm) in the dark. To remove background fluorescence, cells were washed twice with PBS with 5% BSA (300× *g*, 10 min each). After this preparation, anti-CA125 PNPs were added to the cells at a ratio of 2000 PNPs/cell as previously described in [18] and incubated for five minutes at room temperature under continuous mixing conditions (750 rpm). Labelled cells were transferred to 15 mL tubes and centrifuged at 300× *g* for 10 min. After discarding, the supernatant cells were reconstituted in 10 mL of PBS with 5% BSA and centrifuged again for 10 min at 300× *g*. Following this wash step, the cells were reconstituted to a density of 25M cells/mL in PBS with 5% BSA. The microscopy slide was prepared by placing a 9 mm diameter Grace Bio-Labs (Bend, OR, USA) SecureSeal imaging spacer, adding 9.1 µL of the labeled cell solution and sealing it with a coverslip.

### 2.3. Microscope and Image Acquisition

A custom microscope was built for this project by Thorlabs Imaging Systems (Sterling, VA, USA). The microscope was based upon Cerna microscope components (Thorlabs), consisting of a darkfield slide holder equipped with side illumination RGB and White LEDs, an EPI-fluorescence imaging unit, a laser-based autofocusing unit and a SCMOS 8 Mp color camera (4096 × 2678 pixels) (Thorlabs CS985CU). The use of LED side illumination allowed the use of high-resolution high-power optics without oil immersion and alignment. The microscope was fitted with an X-Y optical stage (Thorlabs MLS203-1) and a piezoelectric focusing unit (Thorlabs PFM450E) for both focusing and Z-stacking. Different fluorescent channels were obtained using a Pinkel set (Semrock DA/FI/TR/Cy5-4x-B) with four excitation filters (four bands), one multi-band dichroic mirror and a multi-band emission filter. Cycling through excitation filters and the removal of the emission filter for darkfield imaging were done with two automated filter wheels (Thorlabs FW102C). Fluorescent illumination was achieved with a xenon plasma light source. Autofocusing was performed by inserting a colinear IR laser through the light path and measuring its reflection through a fiber optic circulator. This resulted in the identification of the aqueous interface between the cover slip and the microscopy slide. An offset was then applied to find the equatorial plane of cells located on the cover slip. Custom software was built for the synchronization and automated acquisition of the images.

The microscope automatically acquired 105 fields of view per sample, capturing for each field of view 40 slices of a darkfield image stack and 4-channel fluorescence of the central slice. The acquisition routine is as follows: (a) the operator selects a field of view on the center of the microscopy slide, (b) the microscope moves to the upper-left corner of the fields of view to be acquired, (c) for each field of view the microscope obtains the optimal focal plane, (d) obtains the four color fluorescent images by switching the excitation filters, (e) blocks the fluorescent light and turns on the darkfield illumination, (f) acquires 40 axial image slices by moving the piezoelectric focusing unit between ±10 µm from the optimal focal plane in steps of 0.5 µm, (g) turns off LED illumination, unblocks fluorescent excitation, changes field of view and repeats steps (c) through (g). The microscopy system acquires fluorescent and darkfield images of the 105 fields of view in 58 min. A picture of the microscope, darkfield and fluorescent images of a field of view, a z-stack, and prototype cells are shown in Figure 1. 

### 2.4. Image Processing and Data Analysis

#### 2.4.1. Fluorescent Imaging and Color Compensation

While four-channel fluorescence images were acquired, cells positive for a single fluorochrome may exhibit signal on more than one of those channels. To determine the antibody that generated the fluorescent signal, we performed image color compensation, a process analogous to color compensation in flow cytometry, but using images. For each pixel, we acquired a 12-dimensional information vector, composed of the RGB values of the camera for each of the four excitation channels of the Pinkel set. With a 5 by 12 color compensation matrix, we can reduce the dimensionality to a 5-dimensional vector, where each dimension represents the contribution of each individual fluorochrome. The color compensation matrix is determined by using beads stained with each of the fluorochromes as follows. 

Color compensation beads (OneComp eBeads) were stained with each of the staining antibodies to have samples for each of the fluorochromes. The bead staining protocol is as follows. One drop of OneComp eBeads was placed into an Eppendorf tube. Five μL of the antibody conjugate were then added and mixed by pulse-vortexing followed by an incubation at 2–8 °C for 30 min in the dark. Then, 0.4 mL of Flow Cytometry Staining Buffer (FCSB) (Invitrogen eBioscience) was added, and the sample was centrifuged at 500× *g* for 5 min. The supernatant was discarded and 0.2 mL of FCSB was added. Five microscopy slides were prepared following the procedure described in Section 2.2.

Images of at least 7 beads were obtained from each of the compensation bead slides. Beads were detected in the images through intensity thresholding and their median intensity value was computed for the red, green, and blue channels of the color camera for the four excitation filters. Color compensation was performed as follows: for each bead type there were 4 fluorescent channels, each of them imaged with a color camera with three channels (RGB). Therefore, for each fluorochrome there were a total of 12 measurements. A 12 × 5 matrix M was generated. This matrix, given the quantity of the fluorochromes, provides the intensity of the 12 measurements. However, when we analyze a sample, we are interested in the opposite. Given the 12 measurements, we need to infer the quantity of fluorochromes that generated them. We therefore computed the Moose-Penrose pseudoinverse of the matrix M, resulting in the color compensation matrix MInv. Once a fluorescent image stack was obtained, the contribution of each fluorochrome to each pixel was computed by generating a 12-dimensional vector for each pixel and multiplying it by MInv. 

#### 2.4.2. Cell Finding on CD45 Images

Cells were stained with anti-CD45, a pan-leukocyte marker. We used the color-compensated CD45 image to perform cell detection by standard image processing techniques. First, the image was downscaled to 1/4th of the original. A local threshold, using a window of 351 pixels and an offset of 0.05, was applied to the image, with pixels above such threshold considered as belonging to the cell. We then performed connected component analysis in the binarized image, where only components between 500 and 6000 pixels were retained. A circular Hough transform [19] searched for circles, representative of cells, that have a radius between 15 and 35 pixels and are within the connected components. The circle with the highest Hough score and contained within the bounding box of the connected component was retained and identified as a positive cell event.

#### 2.4.3. Cell Contour Delineation

While the cell finding routine provides a good circular approximation of the contour of the cell, such approximation is significantly affected by other cells in close proximity or if the cell has degraded during storage, transport, or staining. To accurately determine the contour of the cell, we made use of artificial intelligence techniques. The determination of the contour of the cell is analogous to its so-called semantic segmentation. Current deep learning segmentation techniques, such as the UNET [20,21], have excelled in the segmentation of cells under different imaging conditions. We used a UNET segmentation neural network to accurately delineate the contour of the cell. Briefly, we collected 3192 images of 82 × 82 pixels and manually delineated the contour of the cell on each of them. We split such images into three sets: training (50%), validation (25%) and test (25%). Such split was done once and never changed. We trained the model using the adaptive momentum optimization algorithm optimizing the Dice [22] coefficient, analogously to [23]. Training was performed for 150 epochs. No early stopping criteria were used. The model with the best validation performance is stored to perform the computation of the cell contour in subsequent images. The number of epochs was selected to force overfitting to the training data. Please note that we selected the model with the best validation accuracy for analysis; therefore, overfitting to the training set is of no consequence for further data analysis. The Dice score in the test images is 0.971. After training, the model is used to generate the contour of the subsequent cells by thresholding the output of the UNET network at a value of 0.5. Cells smaller than 16 µm^2^ or bigger than 400 µm^2^ were rejected, as they are likely artifacts or cells of no interest.

#### 2.4.4. Cell Lineage Determination

Given the individual contributions of each fluorochrome estimated in the color compensation images, and the contour of the cell determined by the UNET segmentation method, we decided if a cell is positive for a given fluorochrome by measuring the signal intensity within the contour and outside of the contour of the cell. If the average value of the fluorescent signal inside the cell was higher than the average plus twice the standard deviation of the fluorescent signal outside the cell, then the cell was considered positive for such fluorochrome. 

Cells positive for CD45 and only one lineage marker were considered the class representative for such lineage marker: CD19 for B cells, CD56 for NK cells, CD3 for T cells and CD14 for monocytes. Cells positive for multiple markers were excluded from this study except for cells positive for both CD56 and CD14, which were labeled as NK-T cells.

#### 2.4.5. PNP Quantification

PNP quantification on cells followed the methodology of [23]. Briefly, for each cell, we extract a centered 3D region of interest. We detect PNPs in each z-slice using a Laplacian of Gaussian blob detection. PNP signals whose x−y coordinates do not lie within the cell contour determined by the UNET are rejected. 2D PNP signals are clustered to form 3D PNP signals using the DBSCAN clustering algorithm [24]. Clusters having too few or too many 2D PNPs are excluded from further analysis. Each remaining cluster is counted as a PNP, and the central 2D PNP determination is selected as the 3D PNP detection coordinate. The color of such a coordinate is evaluated to discriminate between a single PNP (green), a duplet (yellow) or a triplet of PNPs (red). The quantification of the color is made by dividing the red channel of the image at the 3D detection coordinate by the green channel. If the division is between 0 and 1, then the PNP is considered green, if it is between 1 and 2, it is considered yellow and if it is greater than 2 it is considered red.

#### 2.4.6. Data Aggregation

For each sample and cell subtype, we compute the average number of PNPs per cell. This is performed by assuming that the distribution of PNPs in the cell population follows a Poisson distribution. We compute the lambda parameter of such distribution by computing the average number of PNPs per cell. We also include the average number of PNPs per cell without cell subtyping as reference. 

### 2.5. Statistical Analysis

Since most variables failed Shapiro–Wilk’s normality test, we used the non-parametric Mann–Whitney U-test to establish the statistical significance of the difference in the average number of PNPs per cell subtype for the two populations under study: serous ovarian cancer patients and healthy controls. Pearson correlation coefficients were employed to establish correlations between serum and cell-bound CA125. The Python SciPy (version 1.4.1) and statsmodel (version 0.12.1) libraries were used to compute these statistical tests.

## 3. Results

### 3.1. Cohort

The average age of healthy controls was 43.71 years (±10.79). The average age of serous invasive cancer patients was 64.64 (±11.07). All patients were post-menopausal. Five healthy controls were pre-menopausal and 2 post-menopausal. The average serum CA125 value for 12 serous invasive patients was 429 I.U./mL, with a standard deviation of 505 I.U./mL No serum CA125 values were recorded for healthy controls.

### 3.2. Technical Validation

We compared cell lineage determination with the microscopy platform against the flow cytometry reference standard. PBMC from a healthy control were run in a standard flow cytometer and in our system, using the same fluorochromes. The sample was run three times in the newly developed platform. Gating for the flow cytometry data was performed manually, while it was completely automated in our system. The percentage of cells per class are shown in Table 1. 

We evaluated the performance of the PNP quantification method by comparing the number of PNPs per cell detected automatically against manual readings of the same cells. Forty-nine automatically detected 3D darkfield images of cells were selected at random, 49 more were chosen from cells that contained medium to high numbers of anti-CA125 PNPs bound to their surface. This selection was done to have representative images of all binding patterns. Since high binding cells are a minority, selecting all at random would result in a majority of cells having few or no PNPs attached to them, thus biasing the analysis. The 98 image stacks were annotated by two independent readers using a custom-built software annotation interface. Inter-reader agreement reached a squared Pearson correlation coefficient of r^2^ = 0.893, with an average error of 1.61 PNPs per cell. The automated quantification method achieved a Pearson correlation coefficient against manual readings of r^2^ = 0.752, with an average error of 2.72 PNPs per cell.

### 3.3. Serous Invasive Ovarian Cancer Patients Show an Elevation of CA125 Bound to Immune Cell Subsets Compared to Healthy Controls

We computed the average number of PNPs per immune cell type for 14 serous invasive ovarian cancer patients and 7 healthy controls. The average number of cells per sample was 2471 cells. Using the Mann–Whitney U-test to establish statistical significance, serous invasive ovarian cancer patients compared to healthy controls showed elevated binding on B cells (*p* = 0.024), NK cells (*p* = 0.014) and T cells (*p* = 0.001) and no significant difference on monocytes (*p* = 0.397) or NK-T cells (*p* = 0.109). When comparing all cells regardless of cell lineage, serous invasive ovarian cancer patients showed an elevation of the number of PNPs per cell (*p* = 0.040). Box plots showing the number of PNPs per cell subtype and for all cells combined are shown in Figure 2. Whiskers represent the data range once outliers are excluded. Outliers are detected as datapoints that are below or above 1.5 times the interquartile range beyond the low and high quartiles, respectively. The average number of PNPs per cell for each cell subtype for serous invasive ovarian cancer patients and healthy controls are shown in Table 2. We have further analyzed the percentage of cells of serous invasive ovarian cancer patients that have low, medium, and high nanoparticle binding. We used cutoff thresholds of 5 and 10 nanoparticles per cell. The results are shown in Appendix A.

### 3.4. Immune Cell-Bound CA125 Does Not Correlate to Serum CA125 Levels in Serous Invasive Ovarian Cancer Patients

We investigated a correlation between cell-bound CA125 and serum CA125 in patients. We found no correlation for any cell subtype or for all cells aggregated, as shown in Table 3.

We further determined correlation coefficients between cell-bound CA125 for each pair of cell types, for both patients and healthy controls. The correlation coefficients as well as their *p*-values are shown in Table 3. In healthy controls, only monocytes and B cells show a statistically significant correlation. More statistically significant correlations are found in ovarian cancer patients, where B-cell-bound CA125 correlates statistically significantly to monocyte, T-cell- and NK-cell-bound CA125. T-cell-bound CA125 correlates also significantly with NK-T-cell-bound CA125 and strongly and significantly to NK-cell-bound CA125 (r^2^ = 0.8, *p* < 0.01).

## 4. Discussion

MUC16 has an extensive protein backbone and glycosylation [25]. Therefore, it is an excellent candidate for serving as a ligand for several receptors. Since MUC16 is expressed on the tumor cell surface and is also shed, it is also a good candidate for regulating immune cell interactions [26]. It is with this reasoning that we investigated the immune regulatory properties of this mucin.

In this work, we have developed a fully automated alignment-free microscopy platform with concomitant automated image processing methods which detect and quantify cell-bound CA125 on a leukocyte subset-specific basis in large numbers of cells, something that could not be achieved with existing microscopy or flow-cytometry methods. The high sensitivity of the new microscope comes from three factors: first, plasmonic nanoparticles, when imaged under darkfield conditions are many logs brighter than fluorescent stains; second, PNPs do not photo bleach, therefore we can increase the signal by augmenting excitation intensity; and third, individual PNPs are detected against very low local backgrounds by using high-resolution optics that require neither alignment nor oil immersion. We have validated the cell-lineage estimation capabilities of the platform against flow cytometry, achieving similar counts for T cells, B cells, monocytes and NK-T cells. There is a discrepancy for NK cells. This discrepancy in cell counts may have been introduced by different brightness thresholds for the NK cell subset characterized by low CD56 labeling (CD56dim cells).

The combination of high-power optics (63×) without oil immersion and X-Y and Z scanning mechanisms enables the simultaneous, detailed, three-dimensional analysis of individual cells and the extraction of sound statistics on cell populations. Our current system is able to acquire images from 105 fields of view, representing 2000 cells, in one hour. Extracting statistics from the images takes 1.5 h on a modern computer utilizing prototype Python software. We have recently optimized the prototype software and compressed acquisition and processing time to one hour per sample. This is important for laboratories doing clinical research and a major step toward throughputs that are acceptable for routine clinical testing. 

Elevation of cell-bound CA125 on PBMC of serous invasive cancer patients was already shown in flow cytometry [11] and via microscopy [18]. In this work, we furthered our research in cell-bound CA125 by: (a) improving on our sensitive detection technique that enables reliable quantification of CA125 on patient PBMCs to make it oil-immersion- and alignment-free; (b) coupling such sensitive technique to cell-lineaging fluorescent capacities; and (c) automating slide scanning to analyze large numbers of cells, thus enabling the analysis of low-abundance cell populations. 

We have shown that an elevation of cell-bound CA125 is markedly significant on T cells and NK cells, and significant on B cells as well as all PBMCs together. No significant difference was found for NK-T cells and monocytes. Monocytes are the only phagocytes that we analyzed, and it has been reported in the literature that, depending upon size, surface chemistry modification and time, they are phagocytic for nanoparticles [27]. This mechanism presents a confounding factor and currently prevents drawing conclusions on monocyte surface CA125 binding. The monocyte data reported in this work may include both surface bound and internalized nanoparticles.

While our present dataset size is not adequate for a thorough ROC analysis of serous invasive cancer diagnosis, we took preliminary note that the AUC was 0.92 for T cells, 0.78 for NK cells, 0.74 for B cells, 0.56 for monocytes and 0.38 for NK-T cells. This early result suggests future studies to test the diagnostic value of MUC16 binding on a specific leukocyte subset basis.

Our observation that there is no correlation between serum CA125 and PBMC bound CA125 for any of the subsets or all the cells together highlights the incongruence of CA125 measurements in these two compartments. These data are suggesting that the binding of CA125 to immune cells does not match with the stoichiometry of CA125 found in serum of the same individual. Further studies that explore the binding kinetics and anatomical location for the binding of CA125 to immune cells will be necessary to fully address these important initial observations.

Furthermore, in Table 3 we observed several correlations between cell subsets that are statistically significant. It is tempting to suggest that these additional correlations in the cancer patient population may have some value in developing a novel diagnostic algorithm. However, these current data were obtained from only a limited number of samples and, in the case of the healthy donors, the results were obtained from pre- and post-menopausal women. Our future studies will focus on the analysis of many additional healthy donor and cancer patient samples and closely examine the diagnostic value of these correlations between immune cell subsets. Further development of our novel imaging microscope will facilitate such research. 

Based on our prior glycan analysis, it is clear that MUC16 glycans are often terminated with alpha-2, 3-linked sialic acids [28]. These sialic acid residues are recognized by the lectin receptor Siglec-9 [11]. The expression levels of Siglec-9 will affect the binding equilibrium between serum CA125 and cell surface CA125. Further elucidation of Siglec-9 expression in disease could provide insight into the complex relationship between both compartments for CA125. 

The observation that MUC16 is present on T cells is interesting, because recently it has been shown that Siglec-9 is overexpressed on T cells from the ascites and tumor-invading lymphocytes (TILs) of ovarian [29] and other tumors, such as head and neck tumors [30] or acute myeloid leukemia [31]. 

In this work, we have focused on the analysis of T cells, B cells, NK cells and NK-T cells and monocytes by using specific cell-lineage markers. We have mitigated against granulocytes as a possible source of contamination by the use of fluorescence lineage marking to identify mononuclear cell subsets. The CD markers we are using for T cells and B cells protect the results from that contamination, as CD19 and CD3 are not expressed in granulocytes. The marker for NK cells, CD56, is expressed in granulocytes, but only in less than 2% of them, minimizing the effect of contamination. CD14, our marker for monocytes, is weakly expressed in certain granulocyte subpopulations and no difference in PNP binding to monocytes has been found between healthy subjects and controls. 

One limitation of this work is the selection of the healthy controls for comparison with serous invasive cancer patients. Our control group included pre- (N = 5) and postmenopausal (N = 2) women. It is well established that serum CA125 levels in premenopausal women are generally higher than in postmenopausal subjects [32]. The inclusion of premenopausal control samples has the potential to skew the results to higher binding of CA125 to the immune cells, and therefore one might argue that it reduces the differential between the control and the ovarian cancer cohorts. However, we have shown in patient samples that there is no correlation between serum CA125 and cell-bound CA125. On the other hand, should there be a positive correlation in younger healthy women, then the statistically significant differences in cell-bound CA125 levels we observe are even more impressive. Moreover, in our previous study [18], we used a small cohort of five patients and five age-matched controls to show elevation of cell-bound CA125 in patients. Despite the lack of age-matching in the subjects of the present study, the data obtained are coherent with our previous publication. Our future studies will focus on careful recruitment of the subjects to control and test cohorts to obtain a more clinically relevant comparison of immune cell-bound CA125 levels.

## 5. Conclusions

Quantitative measurements of low-abundance cell surface molecules are difficult to perform using standard cytometric techniques. In prior work, we have demonstrated the use of antibody-conjugated plasmonic nanoparticles and darkfield microscopy to perform such measurements on PBMCs subtypes. By incorporating automation into the platform, we are now able to analyze an order of magnitude higher number of cells per patient sample, thus enabling better statistical characterization of low-abundance cell surface molecules on low-abundance cell subtypes. This fully automated microscopy system now enables new studies to examine the relevance of such cell-surface molecules in the course of disease. Focusing on ovarian cancer, we demonstrate increased binding of CA125 to aggregate lymphocyte populations and to B cell, T cell and NK cell subsets of ovarian cancer patients as compared to healthy controls. It is of particular interest that we found no correlation between the abundance of cell-bound CA125 and the level of serum CA125. This observation suggests that serum and the surface of certain white cell subsets are independent compartments for the CA125 biomarker. The new microscope and the data analysis package detailed in this study is now poised to be used for testing a novel diagnostic modality for detection and monitoring of ovarian cancer that uses the circulating immune cells instead of the serum or plasma as the compartment for biomarker detection. Further, this technology is not constrained to ovarian cancer, but can be readily adapted to the quantification of other cell surface molecules that are relevant in the course of disease, especially those associated with cell-mediated immunity.

## Figures and Tables

**Figure 1 cancers-13-02072-f001:**
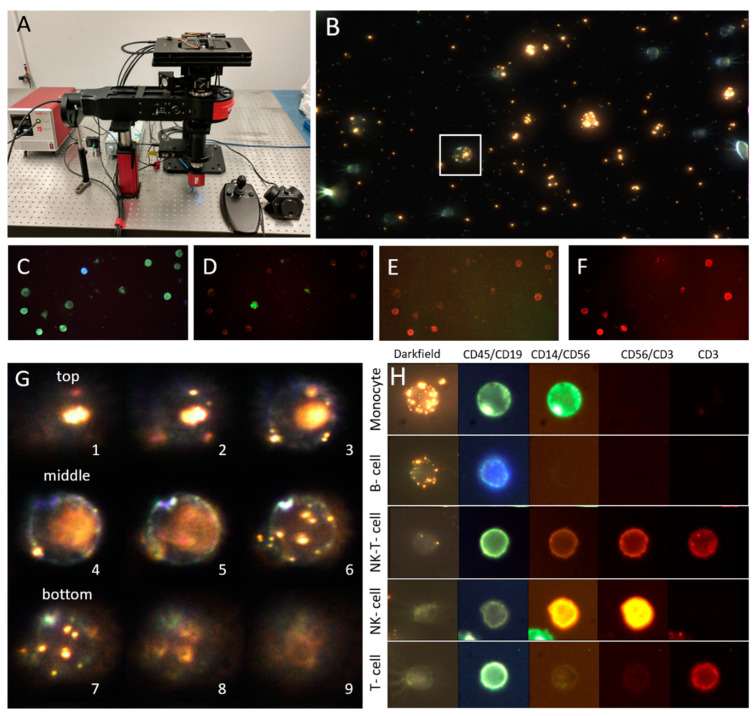
Microscopy system and example images of PBMC from a healthy control. (**A**) Combined automated darkfield and epi-fluorescence microscope developed for the project. (**B**) Maximum-intensity projection of the z-stack darkfield image of a field of view. Some cells show PNPs bound to them. (**C**) First fluorescent channel showing all PBMC in green (CD45) and B cells in blue (CD19). (**D**) Second fluorescent channel showing monocytes in green (CD14) and NK cells in orange (CD56). (**E**) Third fluorescent channel showing NK cells in orange (CD56) and T cells in red (CD3). (**F**) Fourth fluorescent channel showing T cells in red (CD3). (**G**) Representative slices of the z-stack of the monocyte highlighted in B. Please observe how PNPs are visible on the planes of the cell that are above (top) and below (bottom) its equatorial plane (middle), located in the center of the inset. This is due to the high brightness of the plasmonic nanoparticles and the transparency of the cells under darkfield. Numbers indicate the order of the image in the z-stack. (**H**) Composite image showing the darkfield maximum-intensity projection and 4-channel fluorescence of example cells of different lineages. Please observe how, for these examples, the monocyte exhibits high CA125 binding, the B-cell medium CA125 binding and other cell classes show no binding.

**Figure 2 cancers-13-02072-f002:**
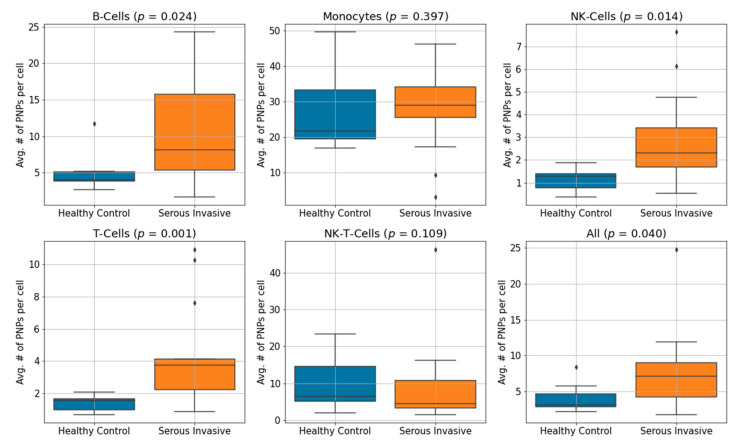
Box plot of the average number of PNPs bound per PBMC subtype for 14 serous invasive cancer patients and 7 healthy controls. *p*-values were obtained using the non-parametric Mann–Whitney U-test. We observe a statistically significant elevation of binding on NK cells, T cells, B cells and all cells. The Y-axis is different between box plots due to the different binding patterns per cell subtype.

**Table 1 cancers-13-02072-t001:** Percentage of cells found by standard flow cytometry and by our system.

	T Cells	B Cells	NK-T Cells	Monocytes	NK Cells
Flow cytometry	74.8%	11.9%	7.14%	6.27%	4.52%
New platform (std)	77.2% (6.7)	7.7% (4.9)	6.2% (1.9)	6.4% (0.7)	1.2% (0.2)

**Table 2 cancers-13-02072-t002:** Average number of PNPs bound per cell subtype for serous invasive ovarian cancer patients and healthy controls. *P*-values are computed using the non-parametric Mann–Whitney U-test.

	All PBMC	NK Cells	T Cells	Monocytes	B Cells	NK-T Cells
Controls	4.11	1.11	1.38	27.58	5.19	10.03
Patients	7.78	2.86	4.36	27.71	10.45	9.10
*p*-value	0.04	0.014	0.001	0.400	0.024	0.109

**Table 3 cancers-13-02072-t003:** Correlation coefficients (r^2^) between the average number of PNPs bound to each leukocyte subtype on 14 serous ovarian cancer patients and 7 healthy controls. The last column represents the correlation to Serum CA125 for the 12 patients with available serum data. Highlighted in bold are statistically significant correlations (*p* < 0.05).

		NK CA125	T CA125	Monocyte CA125	NK-T-CA125	Serum CA125 *
Patients	B CA125	**0.67 (<0.01)**	**0.68 (<0.01)**	**0.32 (0.03)**	0.22 (0.09)	0.07 (0.42)
	NK CA125		**0.80 (<0.01)**	0.18 (0.12)	0.14 (0.19)	0.08 (0.39)
	T CA125			0.23 (0.08)	**0.49 (<0.01)**	0.03 (0.62)
	Monocyte CA125				0.26 (0.06)	0.06 (0.44)
	NK-T-CA125					0.08 (0.36)
	All PBMCs CA125					0.01 (0.81)
Healthy Controls	B CA125	0.17 (0.35)	0.45 (0.10)	**0.71 (0.02)**	0.12 (0.50)	
	NK CA125		0.31 (0.19)	0.03 (0.71)	0.18 (0.39)	
	T CA125			0.48 (0.09)	0.06 (0.64)	
	Monocyte CA125				0.01 (0.84)	

* Serum CA125 data were available for only 12 patients.

## Data Availability

The data used to generate the graphs and perform the data analyses are available upon request to the authors of the manuscript.

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
