# Peer review of "Characterization of Cell-Bound CA125 on Immune Cell Subtypes of Ovarian Cancer Patients Using a Novel Imaging Platform"

_cancers, 2021, doi:10.3390/cancers13092072_

Round 1

Reviewer 1 Report

The major scientific results are well presented and novel for the field, so the article can be accepted after the minor revision. The article text, however, contains some incomplete and non-parallel sentences and a number of possible typos that require some editing. Also I would suggest to add more results to the conclusions section, it is too brief summary of the findings of the article and probably can be extended by presenting not only novelty of the research.

Author Response

We thank you very much for the opportunity to resubmit our manuscript. We also take this opportunity to thank the editors and the three reviewers for their careful reading of the manuscript and the constructive and insightful critique. We have made every attempt to fully address all of the critique and have accordingly modified the manuscript. A point-by-point response to the critique from the reviewers is provided below.

Reviewer 1: Minor

Comment. The major scientific results are well presented and novel for the field, so the article can be accepted after the minor revision. The article text, however, contains some incomplete and non-parallel sentences and a number of possible typos that require some editing. Also I would suggest to add more results to the conclusions section, it is too brief summary of the findings of the article and probably can be extended by presenting not only novelty of the research.

Response. We thank the reviewer for the valuable comments. We have improved the grammar of the text and found several typos. We have also enhanced the conclusions section as suggested, given more context to the findings, and included an enhanced description of our results.

Reviewer 2 Report

The work of Gonzalez and colleagues: “Characterization of cell-bound CA125 on immune cell subtypes of ovarian cancer patients using a novel imaging platform.” Describes the employment of a dark-field (DF) based microscopy for the identification of a novel potential ovarian cancer biomarker – immune-cell bound CA125.

General and major concerns:

Unfortunately, I cannot recommend the presented work for publication in Cancers in its current form. However, if the authors would like to answer some questions and improve the quality of the manuscript correcting certain issues, the manuscript can be re-considered for publication.

  • Please, provide absolutely clear statement what is the novelty of your current work in the comparison with the work of Jeong and colleagues, ACS Sensors, 2020, 5, pp2772-2782. At least the imaging platform, which was used in that manuscript and the strategy of analysis (used gold plasmonic nano-particles and DF microscopy) are very similar. In this case, why the authors mention in the title that platform is “novel”, if this platform was already described by the same working team just few months earlier in ACS Sensors?
  • In the presented study, the control group and the patient group matched neither on age nor on physiological status. In this case, how they may be compared? It is really necessary to have matching groups for such type studies.
  • The authors worked with PBMCs, which were frozen and then recovered. Moreover, PBMC isolation was performed using only Ficoll plaque gradient centrifugation. However, the authors did not provide any information regarding the purity of isolated cells, regarding the viability of recovered PBMCs after thawing. Also the fact that freezing-thawing process does not affect the expression of cell surface markers is not confirmed. Please, address this issue.
  • Please, double-check English for typos.

Even if the Researcher is “private”, the contact information, at least the email of this researcher has to be provided.

Author Response

We thank you very much for the opportunity to resubmit our manuscript. We also take this opportunity to thank the editors and the three reviewers for their careful reading of the manuscript and the constructive and insightful critique. We have made every attempt to fully address all of the critique and have accordingly modified the manuscript. A point-by-point response to the critique from the reviewers is provided below.

Reviewer 2: Major Concerns

Comment 1. Please, provide absolutely clear statement what is the novelty of your current work in the comparison with the work of Jeong and colleagues, ACS Sensors, 2020, 5, pp2772-2782. At least the imaging platform, which was used in that manuscript and the strategy of analysis (used gold plasmonic nano-particles and DF microscopy) are very similar. In this case, why the authors mention in the title that platform is “novel”, if this platform was already described by the same working team just few months earlier in ACS Sensors?

Response 1. The data in this manuscript were acquired on a completely revamped imaging system than the one that was used in our ACS sensors publication. In the new system, we now have the capability to monitor fluorescence in addition to surface plasmon resonance. This was done to allow us to measure CA125 levels on specific leukocyte subsets rather than the whole peripheral blood mononuclear cells (PBMCs) population. As a result, we are now able to incorporate immune subset-specific CA125 binding as additional parameters to distinguish between PBMC samples from ovarian cancer patients and healthy donors.

Secondly, operating our previous technology platform involved complex and tedious user operations, which limited the analysis to 150 cells per sample. This precluded the analysis of the less abundant immune cell subsets such as the NK cells. By completely automating the platform through autofocusing, auto-scanning, automated cell finding and lineage identification we have now developed a system that allows us to analyze 2500 cells per sample and detect increase in binding in less common subsets such as NK cells. The system used in the current study is therefore novel and distinct from the platform that we used in our previous ACS Sensors paper.

Comment 2. In the presented study, the control group and the patient group matched neither on age nor on physiological status. In this case, how they may be compared? It is really necessary to have matching groups for such type studies.

Response 2. We fully agree with the reviewer on the need to compare appropriate control and test subjects. We would like to point out that the current study is the first demonstration that quantification of CA125 binding to immune cell subsets using a highly sensitive imaging system as the one that we have developed, will result in a novel biomarker assay for detection of ovarian cancer. In that respect, this current manuscript should be regarded as a proof-of-concept study that provides the foundation to conduct more detailed clinical studies to validate this novel diagnostic paradigm for ovarian cancer. We plan to conduct a multi-institutional clinical trial where CA125 binding to immune cell subsets will be compared in patients with ovarian cancer, benign gynecologic tumors, endometriosis, liver cirrhosis and other conditions that are associated with increased serum CA125 levels in addition to healthy women from both the pre- and post-menopausal groups.

We do acknowledge the importance of the point raised by the reviewer and have modified the last paragraph in Discussion to more fully address this limitation of our study. The last paragraph of Discussion now reads as follows:

“One limitation of this work is the selection of the healthy controls for comparison with serous invasive cancer patients. Our control group included pre- (N=5) and postmenopausal (N=2) women. It is well established that serum CA125 levels in premenopausal women are generally higher than in postmenopausal subjects [31]. The inclusion of premenopausal control samples has the potential to skew the results to higher binding of CA125 to the immune cells and therefore one might argue that it reduces the differential between the control and the ovarian cancer cohorts. However, we have shown in patient samples that there is no correlation between serum CA125 and cell-bound CA125. On the other hand, should there be a positive correlation in younger healthy women, then the statistically significant differences in cell-bound CA125 levels we observe are even more impressive. Moreover, in our previous study [18], we used a small cohort of five patients and five age-matched controls to show elevation of cell-bound CA125 in patients. Despite the lack of age-matching in the subjects of the present study, the data obtained is coherent with our previous publication. Our future studies will focus on careful recruitment of the subjects to control and test cohorts to obtain a more clinically relevant comparison of immune cell bound CA125 levels.”

Comment 3. The authors worked with PBMCs, which were frozen and then recovered. Moreover, PBMC isolation was performed using only Ficoll plaque gradient centrifugation. However, the authors did not provide any information regarding the purity of isolated cells, regarding the viability of recovered PBMCs after thawing. Also the fact that freezing-thawing process does not affect the expression of cell surface markers is not confirmed. Please, address this issue.

Response 3. We thank the reviewer for these valuable comments. We would like to address them point by point

While it is true that we have not assessed the purity of the PBMCs obtained by Ficoll plaque gradient centrifugation in this current study. However, our group has conducted and reported extensive phenotypic and functional assays on the frozen PBMC samples from healthy donors and ovarian cancer patients (please see references [1-4] listed below). We routinely find that the viability of the frozen PBMC after thawing is >80%. The frozen PBMC samples used in the current manuscript were cryopreserved and thawed using the same established protocols in our laboratory and we therefore do not expect the results to be affected due to the quality of the PBMCs. Additionally, we used morphological integrity as a criterion for accepting a sample for analysis. Samples with a high percentage (>25%) of ruptured cells or with large aggregates cells were rejected. We believe our prior experience and the morphology quality check included in our analysis provides the safeguards to ensure that only data from viable cells was included in this study.

The concern about contamination can be fully addressed by considering two important factors. First, the number of ovarian tumor cells is miniscule in blood as evidenced by the extremely low number of cancer cells found in several reports on circulating tumor cells in the blood of patients with ovarian and other cancers.

Second, it is also important to point out that the fluorescence imaging capability of our system allows us to rule out contamination and provides an accurate identification of the PBMC subsets. For example, by including CD45 in our fluorescent antibody panel, we are able to successfully exclude non-immune cells from our analysis.

The CD markers we are using for T-Cells and B-Cells protect the results from falsely identifying granulocytes since these immune cells do not express CD3 and CD19. The marker for NK cells, CD56, is expressed in granulocytes, but only in less than 2% of them, minimizing the effect of contamination. CD14, our marker for monocytes, is weakly expressed in certain granulocyte subpopulations and no difference in PNP binding to monocytes has been found between healthy subjects and controls. Given this analysis, should there be granulocytes into the PBMC preparation, they would most likely be positive for CD45 and negative for our lineage markers. They would therefore be excluded from the analysis. We have added a paragraph to the discussion section reflecting this analysis.

With respect to the effect of the freezing-thawing cycle in our experiments, we would like to clarify that CA125 is not a molecule expressed by leukocytes, but is part of the MUC16 glycoprotein that binds to the siglec-9 receptor of leukocytes [2]. We have not yet analyzed the effect of storage time and temperature of the frozen PBMCs, but plan to do it in future studies. We used a thawing process that is well described in the literature and did not exceed any of the limits set in [5]. Freezing and thawing may indeed have effects on the levels of MUC16 that we observe. Since MUC16 is bound to the siglec-9 receptor on all cell types, a decreased binding level would be consistent across all cell types and would not introduce a differential bias. In addition, all samples followed the same freezing/thawing protocol, therefore the bias, if any, introduced by the freezing/thawing procedure should be consistent for all samples and the differential measurements across cohorts should still be valid.

[1].       Vazquez J, Chavarria M, Lopez GE, Felder MA, Kapur A, Romo Chavez A, Karst N, Barroilhet L, Patankar MS, Stanic AK. Identification of unique clusters of T, dendritic, and innate lymphoid cells in the peritoneal fluid of ovarian cancer patients. Am J Reprod Immunol. 2020;84(3):e13284. Epub 2020/06/12. doi: 10.1111/aji.13284. PubMed PMID: 32524661; PMCID: PMC7754790.

[2].       Belisle JA, Horibata S, Jennifer GA, Petrie S, Kapur A, Andre S, Gabius HJ, Rancourt C, Connor J, Paulson JC, Patankar MS. Identification of Siglec-9 as the receptor for MUC16 on human NK cells, B cells, and monocytes. Mol Cancer. 2010;9:118. Epub 2010/05/26. doi: 10.1186/1476-4598-9-118. PubMed PMID: 20497550; PMCID: PMC2890604.

[3].       Belisle JA, Gubbels JA, Raphael CA, Migneault M, Rancourt C, Connor JP, Patankar MS. Peritoneal natural killer cells from epithelial ovarian cancer patients show an altered phenotype and bind to the tumour marker MUC16 (CA125). Immunology. 2007;122(3):418-29. Epub 2007/07/10. doi: 10.1111/j.1365-2567.2007.02660.x. PubMed PMID: 17617155; PMCID: PMC2266014.

[4].       Felder M, Kapur A, Rakhmilevich AL, Qu X, Sondel PM, Gillies SD, Connor J, Patankar MS. MUC16 suppresses human and murine innate immune responses. Gynecol Oncol. 2019;152(3):618-28. Epub 2019/01/11. doi: 10.1016/j.ygyno.2018.12.023. PubMed PMID: 30626487.

[5] Hønge BL, Petersen MS, Olesen R, Møller BK, Erikstrup C (2017) Optimizing recovery of frozen human peripheral blood mononuclear cells for flow cytometry. PLoS ONE 12(11): e0187440. https://doi.org/10.1371/journal.pone.0187440

Comment 4. Please, double-check English for typos.

Response 4. We thank the reviewer for this suggestion. We have improved the grammar of the text and corrected the typographic errors.

Comment 5. Even if the Researcher is “private”, the contact information, at least the email of this researcher has to be provided.

Response 5. We have added the email address of the private researcher.

Reviewer 3 Report

Comments to the authors:

The study by González et al. is a continuation of their previous work on CA125 expression and binding detection in tumor cells and PBMCs, using novel plasmonic gold nanoparticles. Using this interesting method, authors focus upon improving detection sensitivity of low abundance proteins and ligands compared to usual flow cytometry and fluorescent microscopy. In this study, they refine the method described in the previous publication and by combining PNP detection with fluorescent lineage markers attempt to define relative CA125 binding on different PBMC lineage in healthy donors and ovarian cancer patients.

The findings of the publication are very interesting and open many doors for future exploration and studies, but few comments and questions must be addressed by the authors before this manuscript is fully complete for publication purposes.

Major comments:

  1. A question for the authors about the method. In the method section, it is stated that all PNP signals inside the cell contour are counted. Does that mean that any PNPs that are internalized are also included in the count?

    Although this is not a big problem for most of the immune cell subtypes it could give a false signal in the monocyte population. Since the monocytes are professional phagocytes nanoparticles were likely uptaken by them which contributes to unspecific/background PNP signal in this cell type. This could provide an alternative explanation to both healthy donor and OC monocytes having high PNP binding.

    Before authors can discuss possible implications of CA125 binding to monocytes they should perform additional experiments using PNP-biotin, similar to the experiments in their previous publication. In case the monocyte population shows again high PNP signal, this has to be acknowledged as a general limitation of the technology in the discussion and conclusion sections. If the PNP-biotin signal is similar to other immune cell subtypes then the CA125-PNP signal is not a false one, and binding preference would match Siglec-8 expression on immune cell subtypes.

  2. Since the authors mention that if the random selection was used the most of the cells would be low to non-PNP binding, is it possible for them to show results for the random selection acquired cells? In this 'true to ex vivo' approach, what would be the distribution of high, medium, and low PNP binding on immune cell subtypes? From the results of the publication mentioned in Ref.11 wouldn't one expect that immune cell subtype CA125 binding would be higher and present on a lot of cells? Experiments using FACS in that publication show 40-90% of CA125+ cells depending on the immune cell subtype from OC patients. Can authors comment where does this difference in results obtained by FACS and PNP method come from?

Minor comments:

  1. A short sentence or two could be added in the introduction explaining the physical phenomenon that enables the detection of PNP by light microscopy.

  2. In the description of Figure 1. B-F perhaps it would be better to address the markers depicted in the particular channels instead of the cell subtype. On the other side of composite image H, marker positivity could be added as well.

  3. Can authors state what was the percentage of excluded cells in their cell lineage estimation? Is this something that influences the discrepancy between flow cytometry and the novel system results depicted in Table 1? The first Table 1., since in the text there are two Table 1.

  4. Reference [31] is not mentioned in the text at any point. It should be excluded from the list added in the text. Please recheck the reference list.

  5. What exactly is correlated in Table 2.? If it is the average number of PNPs per cell subtype then it would be more appropriate to write it like that.

  6. To further establish the relevance of these findings in diagnostics/prognostics authors could attempt using ROC analysis in which they would correlate the average number of bound PNPs to a particular immune cell subtype to YES-NO question (Progression, recurrence, healthy donor vs OC patient). It would indicate the possibility of PNP use as a medical tool. The authors could consider it also for future studies.

Author Response

We thank you very much for the opportunity to resubmit our manuscript. We also take this opportunity to thank the editors and the three reviewers for their careful reading of the manuscript and the constructive and insightful critique. We have made every attempt to fully address all of the critique and have accordingly modified the manuscript. A point-by-point response to the critique from the reviewers is provided below.

Reviewer 3: Major comments

Comment 1. A question for the authors about the method. In the method section, it is stated that all PNP signals inside the cell contour are counted. Does that mean that any PNPs that are internalized are also included in the count? Although this is not a big problem for most of the immune cell subtypes it could give a false signal in the monocyte population. Since the monocytes are professional phagocytes nanoparticles were likely uptaken by them which contributes to unspecific/background PNP signal in this cell type. This could provide an alternative explanation to both healthy donor and OC monocytes having high PNP binding.

Before authors can discuss possible implications of CA125 binding to monocytes they should perform additional experiments using PNP-biotin, similar to the experiments in their previous publication. In case the monocyte population shows again high PNP signal, this has to be acknowledged as a general limitation of the technology in the discussion and conclusion sections. If the PNP-biotin signal is similar to other immune cell subtypes then the CA125-PNP signal is not a false one, and binding preference would match Siglec-8 expression on immune cell subtypes.

Response 1. We would like to thank the reviewer for this very well-developed question. Phagocytosis of gold nanoparticles by monocytes should be taken into account in the analysis. However, it has been published that the upper size limit for non-chemically induced phagocytosis by monocytes is approximately 10 nm [1]. Our nanoparticles are highly monodisperse at 80 nm ± 5%, well above that limit. Further, our system is capable of the acquisition of 3D images of the entire cell volume, enabling us to estimate PNP phagocytosis. We have not seen nanoparticles in the interior of monocytes. As an example, we can refer to Figure 1 G, which is a monocyte imaged at different z-planes under our darkfield microscope. The equator of the cell is depicted in the first two images of the second row. It is clear that no PNP lies inside the equatorial plane of the cell. In this cell, PNPs appear on the “north pole”, the “south pole” and its equator. 

Alternatively, the reason for high PNP binding on monocytes can be found in the high expression level of the receptor Siglec-9 on these cells. Siglec-9 is expressed by >95% of monocytes [2] and, when measured with flow cytometry, the presence of Siglec-9 on their surface is an order of magnitude greater than that of B-Cells or NK-Cells [2, figure 3]. Our data is consistent with these numbers. PNP binding to monocytes is 20-fold greater than PNP binding to NK-Cells and 5-fold greater than binding to B-cells.

[1] Gustafson HH, Holt-Casper D, Grainger DW, Ghandehari H. Nanoparticle Uptake: The Phagocyte Problem. Nano Today. 2015;10(4):487-510. doi:10.1016/j.nantod.2015.06.006

[2] J. A. Belisle et al., “Identification of Siglec-9 as the receptor for MUC16 on human NK cells, B cells, and monocytes,” Mol. Cancer, vol. 9, no. 1, p. 118, May 2010, doi: 10.1186/1476-4598-9-118.

Comment 2. Since the authors mention that if the random selection was used the most of the cells would be low to non-PNP binding, is it possible for them to show results for the random selection acquired cells? In this 'true to ex vivo' approach, what would be the distribution of high, medium, and low PNP binding on immune cell subtypes? From the results of the publication mentioned in Ref.11 wouldn't one expect that immune cell subtype CA125 binding would be higher and present on a lot of cells? Experiments using FACS in that publication show 40-90% of CA125+ cells depending on the immune cell subtype from OC patients. Can authors comment where does this difference in results obtained by FACS and PNP method come from?

Response 2. We would like to thank the reviewer for this comment and clarify some aspects that might have not been clear in our submission. The reviewer points out that by FACS analysis in reference 11, ovarian cancer patients have 40-90% CA125+ PBMCs, in other words, 40-90% PBMCs were above a cutoff for fluorescence intensity. We would like to point out that such publication found the range to be 2-90%, where the 2% represented the population of positive T-Cells. Reference 11 further states that the FACS method was not sensitive enough to give a reliable result for MUC16 binding to T-Cells. While our method of reporting binding did not use a cutoff, but rather used the average number of cell-bound PNPs to each cell type, we have applied the reference 11 analysis to our data and established a range of 22-93%, where 22% represents NK cells and 93% represents monocytes. Please bear in mind that our study defines NK cells as all PBMCs being CD56 positive, while reference 11 focuses on the analysis of CD16pos/CD56dim NK cells. Please, also bear in mind that we are measuring epitope presence using a more sensitive cytometry method.

To answer the question of the reviewer “what would be the distribution of high, medium, and low PNP binding on immune cell subtypes?”, we have constructed the following table using cutoffs of 5 PNPs and 10 PNPs per cell to delineate low, medium and high binding

Cell type           Low (<5 PNPs cell)        Medium (5 ≤ PNPs/cell <10 )      High (≥ 10 PNPs/cell)

B-Cell               40.66                            7.59                                          51.73

NK-Cell             85.19                            5.21                                            9.59

T-Cell                82.05                            5.59                                          12.35

Monocyte           9.82                            3.96                                          86.21

NK-T-Cell          71.15                            4.83                                          24.00

While having shared these results when cutoffs are applied to our data, we do not wish to adopt the method in our technology. The traditional cutoff technique used in flow cytometry suffers from an important flaw. That is that while two samples can have identical percentage positivity by the cutoff method, they may differ dramatically in the level of cell epitope abundance. By using the method we report on the manuscript, which is reporting the average number of PNPs bound per cell subtype, we are reporting a “concentration” result (number of PNPs per cell) rather than a percent of positive cells above a threshold result. By using the “concentration” method we intend to track the quantitative binding level of MUC16 longitudinally and provide better clinical management of ovarian cancer in patients. We plan to perform such analyses in future studies.

Reviewer 3: Minor comments:

Comment 3. A short sentence or two could be added in the introduction explaining the physical phenomenon that enables the detection of PNP by light microscopy.

Response 3. We thank the reviewer for this comment. We have added two sentences in the introduction describing the plasmon resonance effect.

Comment 4. In the description of Figure 1. B-F perhaps it would be better to address the markers depicted in the particular channels instead of the cell subtype. On the other side of composite image H, marker positivity could be added as well.

Response 4. We agree the figure will be clearer if we included the markers in the caption. We have modified it accordingly. We have also modified the composite image H following the reviewers’ suggestions.

Comment 5. Can authors state what was the percentage of excluded cells in their cell lineage estimation? Is this something that influences the discrepancy between flow cytometry and the novel system results depicted in Table 1? The first Table 1., since in the text there are two Table 1.

Response 5. While the discrepancy in cell counts between flow cytometry and the novel system are low for T-Cells, B-Cells, Monocytes and NK-T-Cells, there is a more marked discrepancy for NK cells. This discrepancy in cell counts may have been introduced by different brightness thresholds for the NK cell subset characterized by low CD56 labeling (CD56dim cells).

Comment 6. Reference [31] is not mentioned in the text at any point. It should be excluded from the list added in the text. Please recheck the reference list.

Response 6. We thank the reviewer for the careful check of the reference list. We found that reference 31 was mentioned in the discussion section. We have checked other references for their presence in the text.

Comment 7. What exactly is correlated in Table 2.? If it is the average number of PNPs per cell subtype then it would be more appropriate to write it like that.

Response 7. We agree with the reviewer that Table 2 was not clear with respect to what was being correlated. We have modified the caption of the table accordingly.

Comment 8. To further establish the relevance of these findings in diagnostics/prognostics authors could attempt using ROC analysis in which they would correlate the average number of bound PNPs to a particular immune cell subtype to YES-NO question (Progression, recurrence, healthy donor vs OC patient). It would indicate the possibility of PNP use as a medical tool. The authors could consider it also for future studies.

Response 8. We thank the reviewer for this suggestion. We will report a careful ROC analysis for individual leucocyte subsets once we gain access to larger patient and control populations. In the meantime, we have added the following statement to the Discussion section of the manuscript.

“While our present data set size is not adequate for a thorough ROC analysis of serous invasive cancer diagnosis, we took preliminary note that the AUC was 0.92 for T-cells, 0.78 for NK cells, 0.74 for B cells, 0.56 for monocytes and 0.38 for NK T cells. This early result suggests future studies to test the diagnostic value of MUC16 binding on a specific leukocyte subset basis.”

Round 2

Reviewer 2 Report

The work of Gonzalez and colleagues: “Characterization of cell-bound CA125 on immune cell subtypes of ovarian cancer patients using a novel imaging platform.” Describes the employment of a dark-field (DF) based microscopy for the identification of a novel potential ovarian cancer biomarker – immune-cell bound CA125.

Minor concerns:

I would like to recommend the work of Gonzalez and colleagues for publication after correction of several minor points. Here they are:

  • Page 1 line 37 (Abstract): please change to “ Differences in CA125 binding….were not significant…”
  • Line 120: please change “RPM” to “rpm”.
  • Line 193. Please omit “by”.
  • Line 426. Current sentence needs correction and may be changed to: “….and detailed investigation the diagnostic value of……”

Reviewer 3 Report

The authors have replied to all reviewer points regarding different critical points of the manuscript. Additionally, the edits they have done have helped the manuscript to gain clarity and accessibility to the readers. Still, a couple of points need to be rectified before the manuscript can be published. The PDF and the Word versions that were uploaded to the MDPI reviewer system look the same. 

  1. The statement regarding the ROC analysis that the authors mentioned is completely missing. Would it be possible for the authors to add the missing ROC statement (Response 8.) to the discussion? Additionally, could the authors add the author response 5. to state a possible reason for NK cell discrepancy.

  2. Even though the cutoff analysis and this type of approach is not the focus of the paper as the authors state, the table authors show to the reviewer serves as an interesting comparison to the standard flow cytometry method and could itself provide information on the heterogeneity of MUC16 binding on the leukocyte subsets. Perhaps this table could be included as a supplementary figure?

Regarding major point 1. 

  1. Could the authors directly quote the statement in which the upper size limit for non-chemically induced phagocytosis by monocytes is approximately 10 nm? This is unlikely. In the part of the review where monocyte phagocytosis is discussed, it is stated that phagocytosis happens in the presence of particles below 10 microns or even more optimally, below 6 microns. With the used NPs being in 80-100 nm size this puts them in a small enough size for the phagocytosis. Although the microscopy picture 1.G would imply that NPs are not internalized the question of unspecific binding remains.

  2. This is a possibility that should be fully addressed in the manuscript before any healthy/patient monocyte conclusion can be made. Authors could perform the experiment with biotin-PNPs vs CA125-PNPs and see the amount of the unspecific NP monocyte binding even in the healthy donor to settle the matter. I am aware of references regarding monocytes and Siglec-9, but to fully be certain this additional experiment would serve as an 'isotype' staining.
  3. Could the authors add the labeling for the cell z-stack (equatorial plane, “north” and “south” pole) in figure 1.G?

Round 3

Reviewer 3 Report

Gonzales and the colleges have performed the necessary corrections and acknowledged all reviewer comments. They have incorporated the changes that increase the clarity of the manuscript and brought the conclusions and discussion in line with the data. Their work has interesting implications for potential ovarian cancer diagnostics and fine quantification of MUC16 bound to the leukocyte surface. This reviewer is excited to see further achievements of the authors and recommends the manuscript for publication in the present form.